# Application of Feature Selection and Deep Learning for Cancer Prediction Using DNA Methylation Markers

**DOI:** 10.3390/genes13091557

**Published:** 2022-08-29

**Authors:** Rahul Gomes, Nijhum Paul, Nichol He, Aaron Francis Huber, Rick J. Jansen

**Affiliations:** 1Department of Computer Science, University of Wisconsin-Eau Claire, 133 Phillips Science Hall, 101 Roosevelt Ave, Eau Claire, WI 54701, USA; 2Department of Public Health, North Dakota State University, 640S Aldevron Tower, 1455 14th Ave N, Fargo, ND 58102, USA; 3Genomics, Phenomics, and Bioinformatics Program, North Dakota State University, 640S Aldevron Tower, 1455 14th Ave N, Fargo, ND 58102, USA; 4Center for Immunization Research and Education (CIRE), North Dakota State University, 640S Aldevron Tower, 1455 14th Ave N, Fargo, ND 58102, USA; 5Center for Diagnostic and Therapeutic Strategies in Pancreatic Cancer, North Dakota State University, 640S Aldevron Tower, 1455 14th Ave N, Fargo, ND 58102, USA

**Keywords:** DNA methylation, deep learning, breast cancer, TCGA

## Abstract

DNA methylation is a process that can affect gene accessibility and therefore gene expression. In this study, a machine learning pipeline is proposed for the prediction of breast cancer and the identification of significant genes that contribute to the prediction. The current study utilized breast cancer methylation data from The Cancer Genome Atlas (TCGA), specifically the TCGA-BRCA dataset. Feature engineering techniques have been utilized to reduce data volume and make deep learning scalable. A comparative analysis of the proposed approach on Illumina 27K and 450K methylation data reveals that deep learning methodologies for cancer prediction can be coupled with feature selection models to enhance prediction accuracy. Prediction using 450K methylation markers can be accomplished in less than 13 s with an accuracy of 98.75%. Of the list of 685 genes in the feature selected 27K dataset, 578 were mapped to Ensemble Gene IDs. This reduced set was significantly (FDR < 0.05) enriched in five biological processes and one molecular function. Of the list of 1572 genes in the feature selected 450K data set, 1290 were mapped to Ensemble Gene IDs. This reduced set was significantly (FDR < 0.05) enriched in 95 biological processes and 17 molecular functions. Seven oncogene/tumor suppressor genes were common between the 27K and 450K feature selected gene sets. These genes were RTN4IP1, MYO18B, ANP32A, BRF1, SETBP1, NTRK1, and IGF2R. Our bioinformatics deep learning workflow, incorporating imputation and data balancing methods, is able to identify important methylation markers related to functionally important genes in breast cancer with high accuracy compared to deep learning or statistical models alone.

## 1. Introduction

DNA methylation is important in cancer development and progression due to its role in silencing tumor suppressor genes or enhancing oncogene expression [1]. It involves adding a methyl group to the cytosine base pair position in the DNA of a living organism. This epigenetic modification has been demonstrated to directly influence gene expression [2]. Methylation data can be generated using high throughput sequencing techniques [3] where a single donor can have over 850,000 detectable methylation markers (CpGs) [4] across the human genome. Newer sequencing technologies now allow the evaluation of methylation at each genome location with whole genome sequence data. These datasets tend to be too large to reasonably parse through manually. Given the high ratio of markers to samples in these cancer datasets, it is necessary to establish a standardized automated framework that is capable of processing such a massive amount of information, reducing prediction bias, and providing researchers a pipeline with access to pre-trained prediction knowledge. Predictions need to have high accuracy, sensitivity, and specificity. This will save training time and facilitate better knowledge discovery across research groups and cancer types.

Deep learning [5,6] (a subset of machine learning) has been one solution to this big data issue [7] that has gained significant momentum due to its ability to extract a meaningful subset of features from these different datasets [8,9] without any preprocessing or feature transformation. While deep learning algorithms excel in prediction, they can also be computationally expensive. That coupled with high dimensional datasets [10] can make accurate training and prediction very challenging. To overcome these obstacles feature selection prior to application of machine learning algorithms can be used [11]. Feature selection algorithms remove redundant and correlated information from big data thereby scaling down memory constraints. Additionally, data imbalance [12] also poses a challenge to machine learning and is a common issue with most datasets. An imbalanced dataset has more records belonging to a particular class than another. For example, a cancer dataset with 90% normal patients and 10% cancer patients can bias the model to predict a patient as normal. All of these are significant problems that affect neural network models applied to genomic datasets including DNA methylation data generated by high throughput sequencing.

There has been a focus in research on investigating methylation information to predict the relationship between specific gene methylation or expression and cancer. The earliest identified research article surveyed to use deep learning on methylation datasets was published in 2016 [13] and shows a deep learning model capable of predicting DNA methylation state from CpG markers using immortalized myelogenous (K562) cells. Since that time, research in this domain has been restricted to small datasets or to certain types of cancers. In a study by Angermuelle et al. [14], DNA and CpG modules from Single Cell Bisulfite Sequencing (scBS-seq) data and Single Cell Reduced Bisulfite Sequencing (scRRBS-seq) data for Mus musculus (house mouse) were trained using deep learning for prediction of methylated states in a cell. The DNA prediction module utilized a CNN with 2 hidden layers to extract features from DNA sequences while the CpG module utilized a Bidirectional Gated Recurrent Unit to extract features from CpG neighborhoods. This study was limited to only 6 human liver cancer cells (HepG2) [15] and mouse embryonic stem cells (ESCs) [16]. The dataset was small, and the CNN only implemented two convolution layers. Researchers in [17] utilized deep learning to extract DNA methylation states from Nanopore sequencing reads and found the prediction accuracy to be better than traditional techniques such as Hidden Markov Models (HMM). Liu et al. [18] utilized machine learning to extract CpG methylation markers for 27 cancer types from a total of 13,526 samples, where 10,140 samples were cancerous and 3386 were normal. The authors utilized t-statistics to extract the top 2000 CpG markers from 485,000 original CpG sites. The chosen markers were further filtered based on Random Forest and only 12 markers were used to train a deep learning model. While this research utilized a much larger dataset, the manual feature extraction process for selecting the top 2000 markers eliminated more than 99% of the CpG sites. Tian et al. [19], used whole genome bisulfite sequencing (WGBS) data of Human ESCs to predict if the input samples were hypo, hyper or mid-methylated. DNA sequences selected for this analysis were fixed at 400 bps and the input data was fed in CNN as a (400 × 4) feature matrix where 4 stood for bases A, T, C, and G. Authors also noted a data imbalance issue where more data was available on hyper-methylation which led to smaller prediction errors compared to hypo and mid-methylation sites.

Based on the previous work in this domain showing limited use of deep learning and feature selection, the objectives of this research was to develop a bioinformatics workflow incorporating both these aspects to select the most important methylation features associated with breast cancer thereby enabling high predictive accuracy but being scalable at the same time. Thereby our workflow maximizes deep learning predictive accuracy while maintaining scalability.

## 2. Materials and Methods

### 2.1. Dataset

To verify the proposed approach, Illumina 27K and 450K datasets were obtained from the Breast Invasive Carcinoma project from GDC Data Portal [20]. The disease types chosen were Ductal and Lobular Neoplasms. A total of 1188 samples were retrieved. Table 1 shows the distribution of methylation data from the Illumina platform.

Our dataset had CpG markers corresponding to samples with no data. Further analysis showed that the percentage of these markers with null values was independent of cancer tissue type (normal or tumor). As machine learning algorithms do not tend to work well with ‘no data’, these markers were either removed or imputed before proceeding further. We selected either removal or imputation based on research in Lena et al. [21] where authors performed age correlations with methylation beta values before and after imputation. found that imputation for missing 20% information would not introduce a significant margin of error and that statistical tests could validate up to 30% of markers with no data imputed.

Analysis of the 27K dataset revealed 2597 CpG markers with null values across all 337 samples. Hence, a cut-off of 80% for a missing CpG marker across all samples was used to remove a specific CpG marker. The remaining 24,981 CpG markers across the 337 samples had a total of 4911 missing values. Four different imputation techniques were used to fill in these missing values namely zero, k-nearest neighbor (KNN), mean, and iterative imputation. During zero imputation, the missing values are replaced with zero. For KNN-imputation, the missing CpG markers are imputed using the average values or weighted by Euclidean distance from CpG markers distributed across ‘k’ neighboring samples [22]. Mean imputation calculates a simple average of CpG markers and assigns that value to the missing CpG marker. Finally, with iterative imputation, the samples with missing CpG markers are imputed by modeling each marker with missing value as a function of other markers in an iterative and round-robin manner. The imputer used Bayesian Ridge regression [23] to draw a probabilistic model for estimating missing values.

A similar path was followed for imputation for the 450K dataset. Initially, there were 485,577 CpG markers with almost 89,671 CpG markers having null values across all 851 samples. However, unlike in the 27K dataset, there were several samples with a significant percentage of null values. Hence, a cut-off of 30% for a missing CpG marker across samples was used to remove them. After removal, the dataset had 395,722 CpG markers across the 851 samples with a total of 332,330 missing values that required imputation. A similar process of mean, zero, and KNN imputation techniques were followed with the exclusion of iterative imputation due to memory constraints. Table 2 shows the error associated with these imputation approaches while Figure 1 shows the missing percentage of methylation values in the samples across datasets. These imputation techniques were verified using the Random Forest [24] regression technique. Based on the lowest MSE results, the mean imputation dataset was used for 27K, and zero imputation was used for 450K.

### 2.2. Feature Selection to Reduce Dimensionality

Our developed workflow is visualized in Figure 2. It begins with the pre-processing steps explained in the methods, the feature selection as discussed in this section above, and the implementation of the deep learning model for prediction. In the dimension reduction step, two processes were used, Analysis of variance (ANOVA) and Random Forest.

ANOVA is a technique that can compare the means of different groups. ANOVA uses F-tests (ratio of variances) to statistically test the equality of means. Larger values represent greater dispersion. Therefore if a specific feature, results in more separation in the means or classes, it will have a higher score. ANOVA offers an advantage over the common T-test which is known to use a repeating set of comparisons among two attributes at a time [25]. ANOVA F-test model was trained separately on 24,981 markers from the 27K dataset and 395,722 markers from the 450K dataset.

Random Forests [24,26] are an ensemble learning method that constructs a multitude of decision trees for classification and regression at training time. For classification tasks, the output of the random forest is the class selected by most trees. For regression tasks, the mean or average prediction of the individual trees are [26]. By contrast, variables with low importance might be omitted from a model, making it simpler and faster to fit and predict.

### 2.3. Handling Data Imbalance

The class imbalance of tumor and normal samples is very high in the datasets as observed in Table 1. Such a high imbalance often results in biased prediction and misleading accuracy. One approach to address this challenge is increasing the observations of the minority class, also known as oversampling. For the model, Synthetic Minority Oversampling Technique (SMOTE) [27] was used for oversampling the minority (normal) samples. SMOTE creates new synthetic samples rather than just duplicating examples from minority classes, as duplicating the examples does not add any new information to the model. This technique works by selecting training data that are close in the feature space (nearest neighbors) and generating a new sample in that feature space like the neighbors. SMOTE was applied after oversampling, to ensure that the percentage of tumor and normal samples in our model are equal and to reduce the bias and misinterpretation.

### 2.4. Deep Learning Application for Cancer Prediction

The proposed deep learning sequential model is built using TensorFlow [28]. Two variants of a sequential deep neural network were implemented based on the size of the dataset. The 27K dataset was classified using a neural network with four hidden layers and an output layer. These hidden layers have 10, 20, 30, and 20 neurons respectively. These neurons are passed through a non-linear ReLU [29] activation function. To prevent overfitting the model, a dropout of 0.25 was used after each hidden layer. Dropouts prevent overfitting by turning off a few neurons at random. Since it is a binary classification, the binary cross-entropy loss function was used to evaluate the performance of the model. This loss function was optimized using Adam [30] optimizer. Since the 450K dataset is significantly larger, two different models were tested. The difference between these models can be found in the number of neurons in the model. The standard version had 10, 20, 30, and 20 neurons respectively in the hidden layers while the extended version had 100, 200, 300, and 200 filters thereby allowing the model to pass more information from the dataset. For future reference, the smaller deep sequential network will be referred to as the base model while the extended network will be referred to as the larger model. The model architecture was chosen for its simplicity, which allows for quick compilation and low computation cost. The datasets are simplistic, which means the problem does not require a complex architecture to produce good results. Each variant of the model is trained for 30 epochs. While training, model weights were updated using loss obtained from a validation dataset. The purpose to monitor validation loss is again related to overfitting. Training on a fixed sample size and tuning the model by monitoring its performance on untrained validation data ensures that the model does not overfit the training set. In addition, the learning rate reduction is applied with a factor of 0.5 if validation accuracy does not improve after 5 epochs, with a minimum learning rate of 0.0001.

Due to the imbalance of outcomes in the datasets, preprocessing is aimed toward generating a random, balanced training dataset where there is an equal number of tumor and normal samples. The following steps were used to ensure this selection:Dataset is separated into positive and negative tumor outcomes.The limiting outcome is randomly separated into two sets containing 70% (for training) and 30% (for testing) of the data.A subset of the non-limiting outcome, equal to 70% of the limiting outcome, is randomly chosen.The two subsets of the two outcomes, equivalent in number, is combined to form the training data.All remaining samples are combined to form the testing data set.Both data sets are randomly shuffled internally.

### 2.5. Gene Set Enrichment Analysis (GSEA)

Gene Set Enrichment Analysis (GSEA) was performed on the genes associated with the reduced set of CpG markers. GSEA identifies sets of genes that are enriched in a particular dataset when compared to a control. GSEA considers all genes in the dataset instead of considering only the subsets of genes with significant changes in gene expression. TCGAbiolinks package and ShinyGo [31] were used for performing GSEA.

### 2.6. Survival Analysis

Cox Proportional Hazards modeling was used to determine significant survival differences based on the 7-gene set expression score. These scores were then summed across each of the seven genes split into tertiles for each sample, so each sample would have one total score. This total expression score was categorized into five groups (<2, 2–3, 3–4, 4–5, 5+) because sample sizes were small for more extreme scores. A survival analysis was performed using this total expression score. We used a log-rank *p*-value < 0.05 to indicate a significant difference likely exists between score categories. We feel that further combining categories would obscure any ability to investigate a potential trend or pattern in the data. In this survival analysis, the categories created were combined already to alleviate any sample size or power issues. We propose that presenting the data with more categories will be more informative and helpful to the reader when considering this data and considering similar analyses with their own datasets.

## 3. Results

### 3.1. Feature Selection to Reduce Dimensionality

Figure 3 highlights the feature selection methods implemented in our analysis of four individual datasets. In ANOVA analysis, the markers with a *p*-value greater than a threshold value (0.05/total features) were removed from the total features. The number of features is reduced to 3704 for 27K and 125,949 for 450K datasets. Similar analysis of datasets after application of SMOTE, creating a balanced dataset, saw 15,483 features being selected for 27K and 260,159 features for the 450K. Results from Random Forests saw a lesser number of markers being selected from both datasets compared to ANOVA. Finally, to incorporate results from both these algorithms, the reduced features obtained from the ANOVA F-test are applied to the random forest model. This approach reduced the number of important markers from 24,981 to 336 for the 27K imbalanced dataset, while the final list of markers for the balanced dataset was 475. For the 450K dataset, it was observed that the number of features reduced from 395,722 to 1044 for imbalanced data and 1445 for the balanced dataset.

### 3.2. Deep Learning Application for Cancer Prediction

To verify the efficacy of the proposed feature selection technique, the baseline deep learning model was applied to three variants of the 27K dataset as highlighted in Table 3. The training sample for this dataset without SMOTE application is heavily biased towards tumor samples. To prevent the bias of the result of the sequential model, a 5-fold cross-validation technique was used each time comparing a subset of tumor samples with normal samples, while maintaining an equal distribution of records. For the datasets without SMOTE application there were 309 tumor samples and 28 normal samples. However, the application of SMOTE resulted in 618 samples with 309 samples each for normal and tumor. This process ensures all models are trained on an equal number of positive and negative outcomes (e.g., normal and tumor samples), preventing bias. Figure 4 and Table A1 highlight the results from the proposed deep learning model on 27K dataset.

Two different models were tested on the original 450K datasets. As mentioned earlier, the difference between the two models is the number of filters per layer of the model. The base model is denoted with filters [10, 20, 30, 20] while the larger model has filters [100, 200, 300, 200]. A total of 851 samples were used for both models where 750 samples were tumor and 101 normal as shown in Table 1. Table 3 summarizes the model training set-up for the 450K dataset. Figure 5 summarizes the performance of different approaches for 450K. These results are a summary of accuracy metrics derived in our analysis from Table A2 for the base model and Table A3 for the larger model. Confusion matrices for both 27K and 450K datasets are also available from a separate testing sample that was not used during the training process and are shown in Table A4. Based on these findings, we can conclude that the original 450K and 27K datasets performed poorly in tumor prediction. In Table A4, it can be observed that without SMOTE application and deep learning, the majority of samples have been predicted as tumors even though they were normal. The filtered dataset performs much more reasonably with accuracy values significantly higher than the original dataset and comparable to the SMOTE application dataset. For example, in Table A3, the average accuracy for the filtered dataset is 91.28% while after SMOTE application it is 98.75%. Results after the application of SMOTE on this dataset were very promising. It is conclusive from these graphs that the majority of models trained on data without feature selection and with data imbalance are heavily biased towards predicting only one outcome. This large variability is shown through the excessive standard deviation bars. It is observed that the model performance was significantly better and consistent across different trials using selected features and more so with the balanced dataset produced by SMOTE.

### 3.3. Gene Set Enrichment Analysis (GSEA)

It is important to know the metastatic potential of primary malignant tissue as it is related to the choice of therapy. Previous studies indicate that sets of gene expression profiles can successfully predict survival [32]. After feature selection, four sets of important CpG markers were obtained. The CpGs with the lowest *p*-values (those associated with breast cancer), were annotated to identify which genes were associated with those CpGs. This gene list was then used to perform enrichment analysis and understand how these genes interact, and to infer the functional impact of the CpGs.

Gene Set Enrichment Analysis (GSEA) was performed on six different gene sets associated with the markers identified in Table 3. These were overall 27K and 450K datasets, 27K and 450K with feature selection but no SMOTE, and 27K and 450K with both feature selection and SMOTE. The EAcomplete tool in TCGA biolinks package in R [33,34,35] was used on these sets to identify classes of genes or proteins that are over-represented using annotations for that gene set. The barplot in Figure 6 and Figure 7 shows canonical pathways significantly overrepresented (enriched) by the DEGs (differentially expressed genes) identified from reduced marker datasets after the application of SMOTE. The most statistically significant canonical pathways identified in the DEGs list are listed according to their p-value corrected FDR (−Log) (colored bars) and the ratio of list genes found in each pathway over the total number of genes in that pathway (red line). Plots corresponding to other sets are shown in Figure A1, Figure A2, Figure A3 and Figure A4.

Additionally, GSEA analysis was performed on the 27K and 450K with SMOTE gene lists using ShinyGo [31]. Of list of 685 genes in the 27K SMOTE set, 578 were mapped to Ensemble Gene IDs as shown in Table 4. This reduced set was significantly (FDR < 0.05) enriched in five biological processes, one molecular function, a P53 signaling pathway, and a network visualization of functional associations. (Figure 6b,c) Of the list of 1572 genes in the 450K SMOTE set, 1290 were mapped to Ensemble Gene IDs as shown in Table 4. This reduced set was significantly (FDR < 0.05) enriched in 95 biological processes and 17 molecular function, cellular senescence pathway, and a network visualization of functional associations (Figure 7c).

To evaluate the enriched gene sets identified above and understand their association with cancer, a comparative analysis was performed between these genes with cancer-related genes from other investigated references, one being the combined gene sets from COSMIC [36] and TSGene [37,38]. As shown in Table 4, the TSGene database contains 1217 tumor suppressor genes and the COSMIC has 2172 oncogenes. After combining these two databases (TSGene + COSMIC), a total of 3326 unique genes were identified. Results identified 55 genes that were common between the 27K ANOVA-RF with SMOTE and the list of combined 3326 genes in the COSMIC and TSGene database. Another 136 genes were common between the 450K ANOVA-RF with SMOTE and the 3326 genes. Of the 55 genes in the 27K SMOTE set the top 10 most significantly (FDR < 0.05) enriched biological processes, molecular function, and pathways, and network are visualized. (Figure 8b,c) Of the 136 genes in the 450K SMOTE set the top 10 most significantly (FDR < 0.05) enriched biological processes, molecular function, and pathways, and network are visualized (Figure 9b,c). Of note is the P53 signaling pathway was identified in both gene sets and the gene network is dominated by signaling and cancer-related processes. A list of seven genes were identified as common between them as shown in Figure 10.

### 3.4. Survival Analysis Using Seven Overlapping Genes

Using the seven genes *RTN4IP1, MYO18B, ANP32A, BRF1, SETBP1, NTRK1, IGF2R* identified as overlapping between the 27K with SMOTE, 450K with SMOTE, and the tumor suppressor/oncogene list, an expression score was calculated. For each of the seven genes, expression values were broken into tertiles for all samples. The lowest tertile received a score of 0, middle tertile score of 0.5, and highest tertile score of 1. A survival analysis was performed using this total expression score. The overall log-rank *p*-value (=0.0027) from the survival analysis indicates a significant difference in survival is observed based on the total expression score (Figure 11). Visually, the largest difference appears between the highest expression score group (5+) and the lowest expression score group (<2) with some irregularity in the middle categories when looking at the trend between increasing score and decreasing survival.

## 4. Discussion

In this paper, we demonstrated that imputing missing data and balancing methylation datasets is an important pre-analysis step in the bioinformatics workflow. Our workflow improves the accuracy of breast cancer case prediction using either 27K or 450K methylation datasets. An important note is that the imputation method selected was different depending on the size of the dataset, however, this difference was minimal. Our workflow consistently identified cell signaling and cancer-related processes as important features in predicting breast cancer cases.

Important pathways identified for genes associated with significant CpG markers were the P53 signaling pathway for 27K and cellular senescence for 450K datasets. There is strong evidence in the literature with over 68 studies linking altered P53 signaling with breast cancer, however, none of these studies have demonstrated altered methylation as a potential reason for P53 signaling disruption as we do with this study. Likewise, strong evidence also exists for the association between cellular senescence and breast cancer with 48 study results returned in PubMed. Although again, the evidence linking methylation alterations with these pathways in breast cancer are lacking.

Biological functions identified for genes associated with significant CpG markers were focused around adhesion for 27K and around post transcription and differentiation of cells for 450K datasets. There are 18 studies related to adhesion biological function and breast cancer in PubMed with two studies providing evidence of a potential role of methylation in this relationship. The study by [41] observes that methylation can alter focal adhesion pathways when MCF-7 cells are exposed to cadmium and then selenium. Kominsky et al. [42] reported greater discohesion with hypermethyaltion of CLDN-7 in breast cancer cell lines but not invasive ductal carcinomas.

When focusing on the oncogene/tumor suppressor gene overlap, important pathways included for 27K and for 450K focused on altered cell signaling and cancer. The overlap between the 27K, 450K, and oncogene/tumor suppressor gene list included seven genes (RTN4IP1, MYO18B, ANP32A, BRF1, SETBP1, NTRK1, IGF2R). This list of seven genes highlights those genes that are known to be important in cancer development and happen to be on both Illumina chips. There is one study by Savci-Heijink et al. [43] observing a relationship between RTN4IP1 gene expression and breast cancer. A study by Koo et al. [44] demonstrated a decrease of BRF2 methylation with exposure to soy isoflavone daidzein in breast cancer cells. A study by Di Emidio et al. [45] linking breast cancer treatment, Cyclophosphamide, to altered methylation of Igf2r in mouse offspring suggests that identification of this gene may be a result of treatment rather than cancer process.

The strengths of this study include a large sample size, established, and standardized pre-processing of the methylation data, using simulation to balance an unbalanced dataset, and imputing missing data. There is great overlap between the 27K and 450K datasets in terms of overlapping site-specific markers (94% of 27K loci appear in the 450K set) and correlation of methylation values $\left(R^{2}=0.95\right)$ [46]. This means that any overlapping genes appearing in both datasets represent an independent validation of those overlapping markers as samples will only have either 27K or 450K data but not both in TCGA. All these components help to improve the accuracy and reliability of our analysis. Methylation markers have been associated with genes based on sequence relationships, however, there are likely to be methylation marker effects in more distant genes or based on spatial relationships formed during cell cycle phases that have not been captured within these datasets. Therefore, we expect that some of our gene enrichment interpretations may be missing or inaccurate. The samples were not grouped based on disease features and methylation is a dynamic process that may fluctuate over time, which limits our ability to determine which methylation changes are responsible for the development of breast cancer vs changes that are the result of the presence of breast cancer.

## 5. Conclusions

We proposed a deep learning framework that can capture the most significant biomarkers responsible for breast cancer. Our model is capable of handling a high volume of data with missing values and class imbalance. We observed that reduced features with a balanced class performed better in predicting outcomes than features with an imbalanced dataset. We also performed Gene Set Enrichment Analysis on the sets of genes reduced by our model. To evaluate the efficacy of our model, we compared the reduced sets of genes with several cancer resources. The results seem to support the notion that deep learning methodologies for cancer prediction can be extended for use in the prediction of different types of cancer which will form the basis of our future work. Incorporating methylation data into this story is important from a public health standpoint providing a potential point of prevention and from a treatment standpoint for a potential point to target the P53 or cellular senescence pathways. Deep learning models can be computationally expensive as shown in this research, but to provide accurate results there is a need to handle more diverse datasets as well as take less time to train. Further research will focus on expanding the training dataset to incorporate other clinical variables in the decision-making process. We will also incorporate tumor sub-types and grade information followed by a web-based API to enhance the efficacy of the proposed approach. The feasibility of label-specific weights while training a deep learning model as an alternative to the application of SMOTE will also be explored to verify if that is a more robust technique compared to generating synthetic data for addressing data imbalance. Our focus on methylation markers rather than gene expression has discovered some novelty in breast cancer-specific markers. In addition, these methylation markers have been annotated to be associated with specific genes based on distance, however, these methylation markers may in fact alter or affect other more distant genes in the genome. To supplement this approach, we will explore modifications in the feature selection so that it can accept microarray gene expression data alongside methylation values. This enables the evaluation of specific genes based on their differentially expressed values and their previous association with a cancer type. Identification of these core genes will further reduce methylation markers that are being analyzed by the deep learning model thereby establishing a more robust and targeted approach. By fulfilling these five limitations, we will continue to develop functionality and test its use to enhance the utility of this workflow for the cancer research community.

## Figures and Tables

**Figure 1 genes-13-01557-f001:**
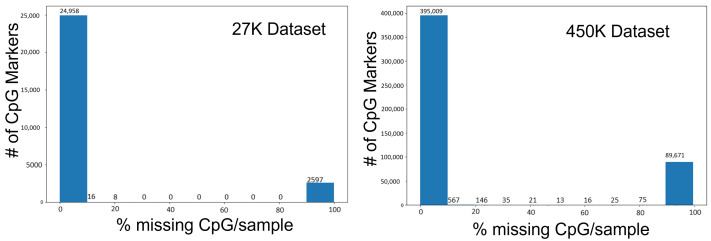
Missing percentage of CpG markers per sample among the 27K and 450K datasets.

**Figure 2 genes-13-01557-f002:**
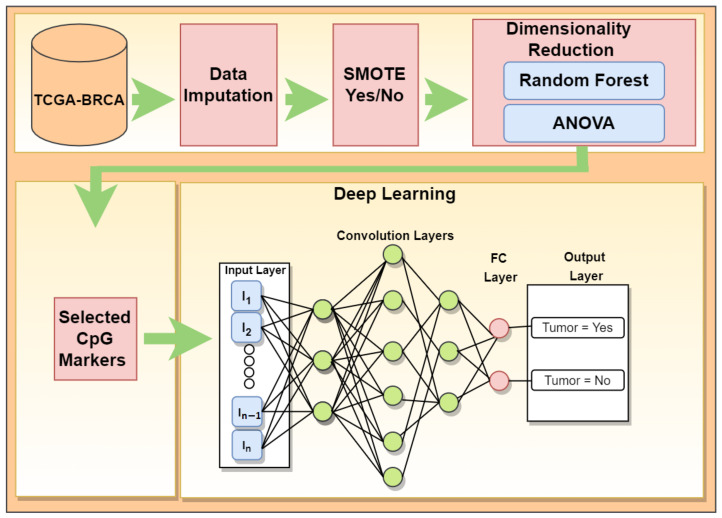
Developed workflow to classify samples as tumor or normal.

**Figure 3 genes-13-01557-f003:**
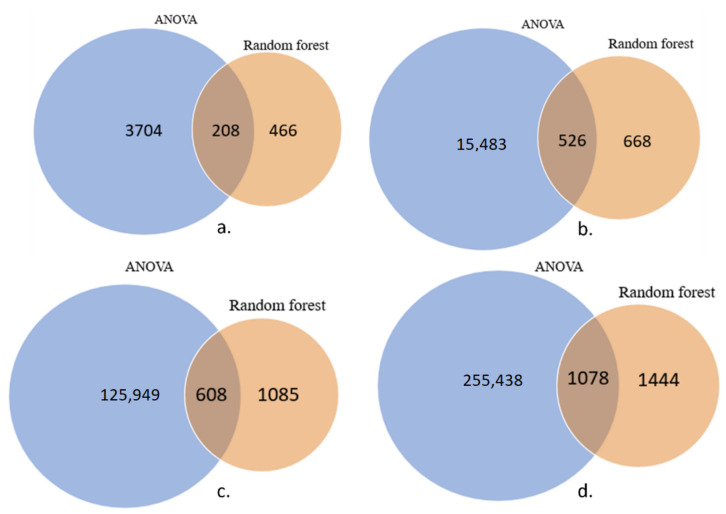
Number of CpG markers selected from (**a**) 27K before SMOTE, (**b**) 27K after SMOTE, (**c**) 450K datasets before SMOTE and (**d**) 450K after SMOTE application.

**Figure 4 genes-13-01557-f004:**
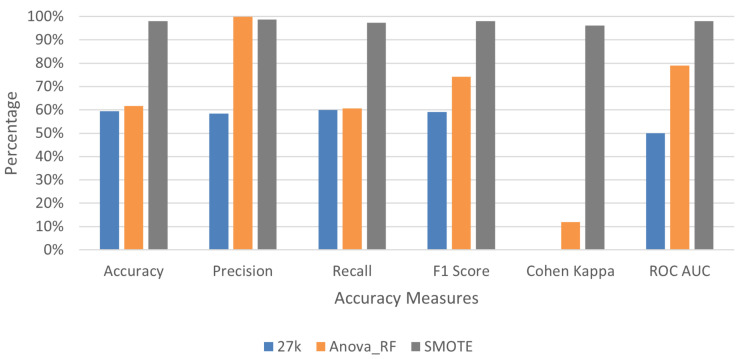
Accuracy metrics from deep learning models on 27K datasets before and after SMOTE application on training data using validation split of 30%. These graphs have been derived from average accuracy values of five trials shown in Table A2.

**Figure 5 genes-13-01557-f005:**
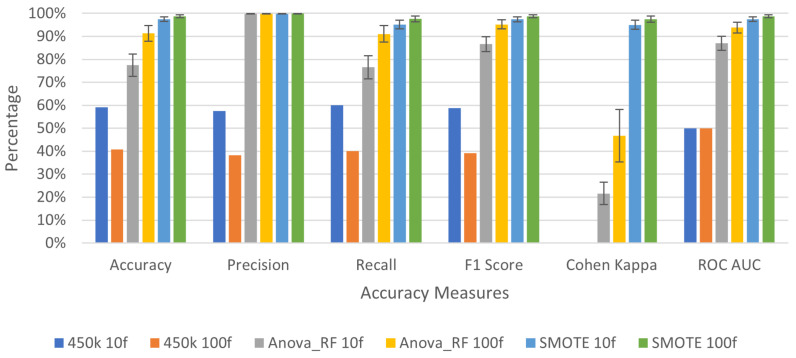
Accuracy metrics from deep learning models on 450K datasets before and after SMOTE application using validation split of 30%. These graphs have been derived from average accuracy values of five trials shown in Table A3.

**Figure 6 genes-13-01557-f006:**
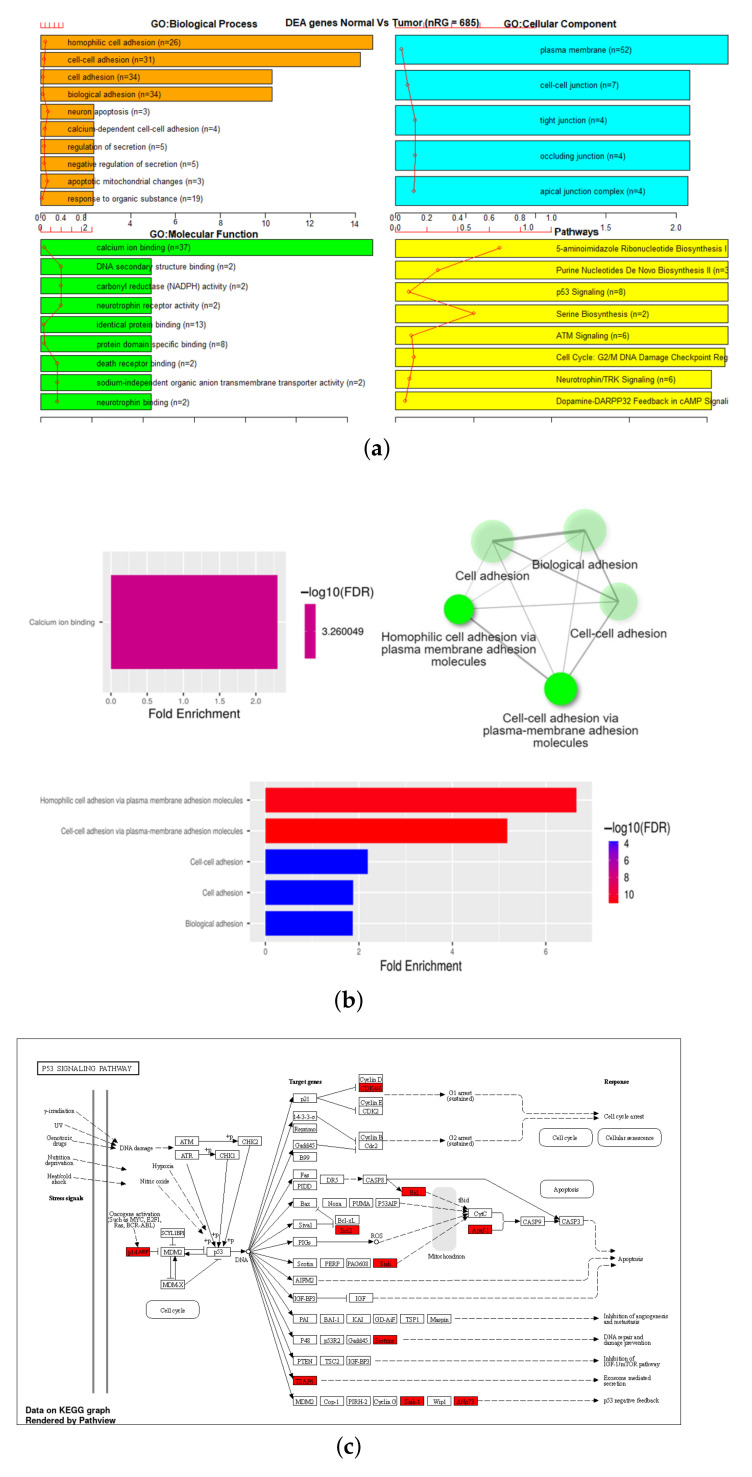
GSEA on reduced markers of 27K SMOTE using (**a**) TCGA EAanalysis where orange represents biological processes, cyan is a cellular component, green is molecular function, and yellow represents pathways, (**b**) ShinyGO with network visualization of functional associations where maroon color represents a molecular function, green represents network-based interaction of biological processes, while the graph in red and blue color represents the strength of the molecular functions identified and (**c**) ShinyGO enriched pathway visualization where red genes are in the gene set [39,40].

**Figure 7 genes-13-01557-f007:**
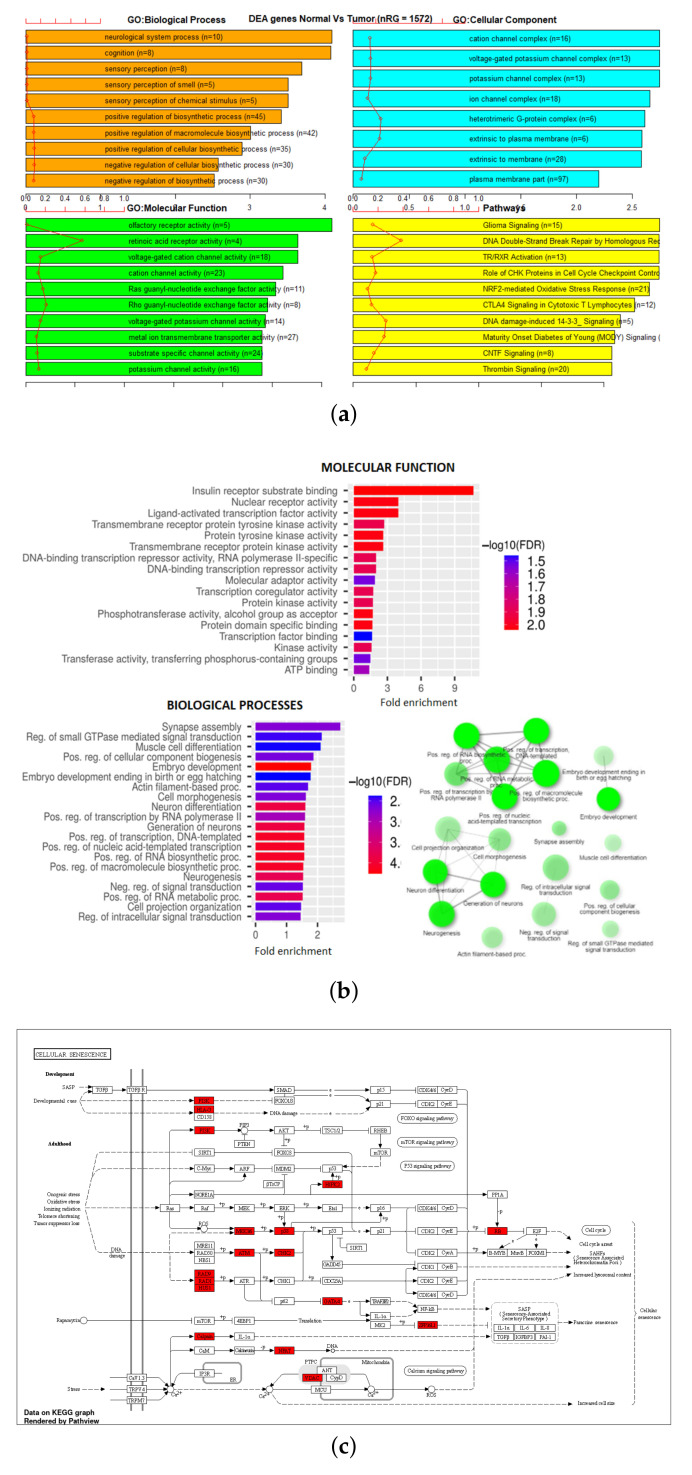
GSEA on reduced markers of 450K SMOTE using (**a**) TCGA EAanalysis where orange represents biological processes, cyan is a cellular component, green is molecular functions, and yellow represents pathways, (**b**) ShinyGO derived biological processes and molecular functions and network-based interaction of biological processes in green nodes, and (**c**) ShinyGO enriched pathway visualization where red genes are in the gene set [39,40].

**Figure 8 genes-13-01557-f008:**
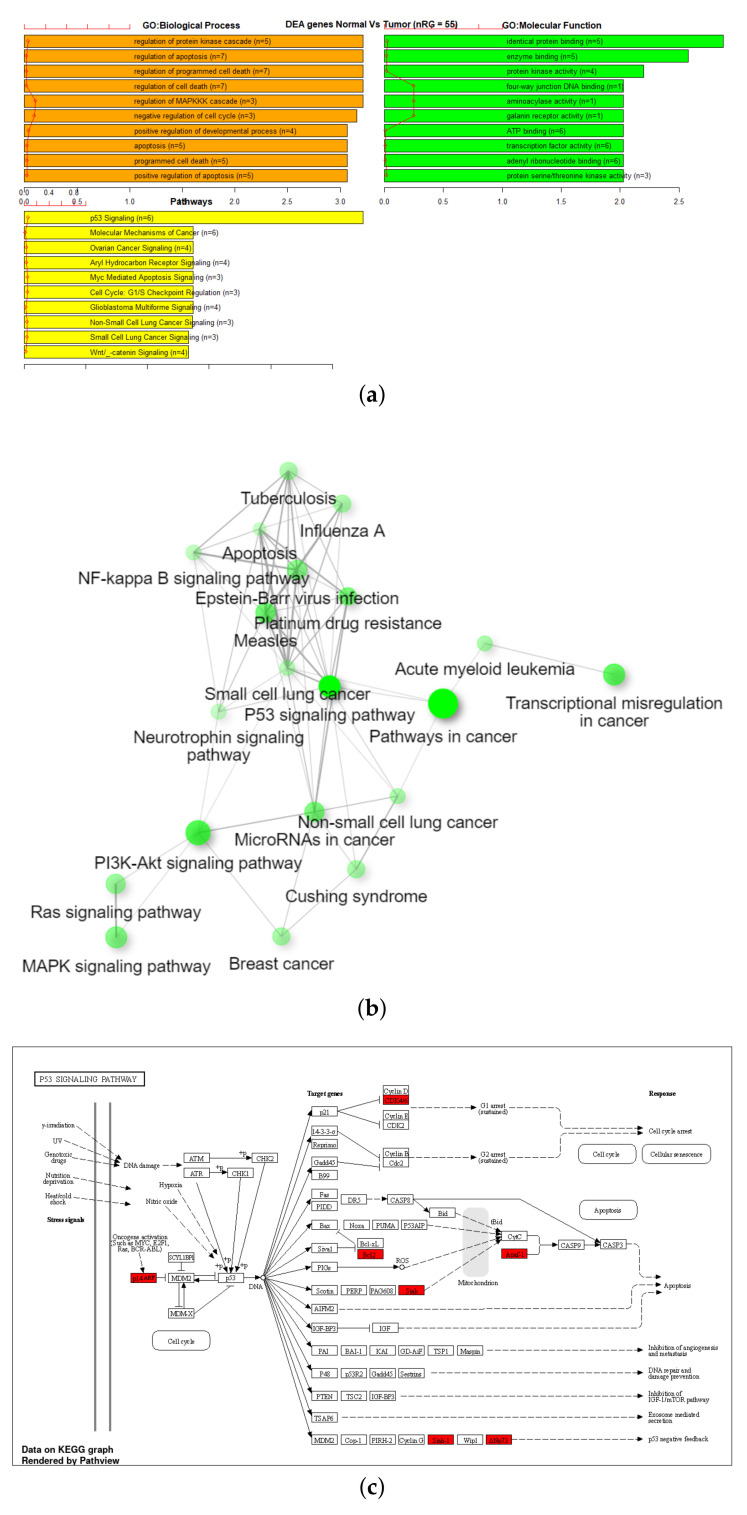
GSEA on Tumor suppressor/oncogene overlap subset of 27K with SMOTE, using (**a**) TCGA EAanalysis where orange represents biological processes, green is molecular functions, and yellow represents pathways, (**b**) ShinyGO with network visualization of functional associations, and (**c**) ShinyGO enriched pathway visualization where red genes are in the gene set [39,40].

**Figure 9 genes-13-01557-f009:**
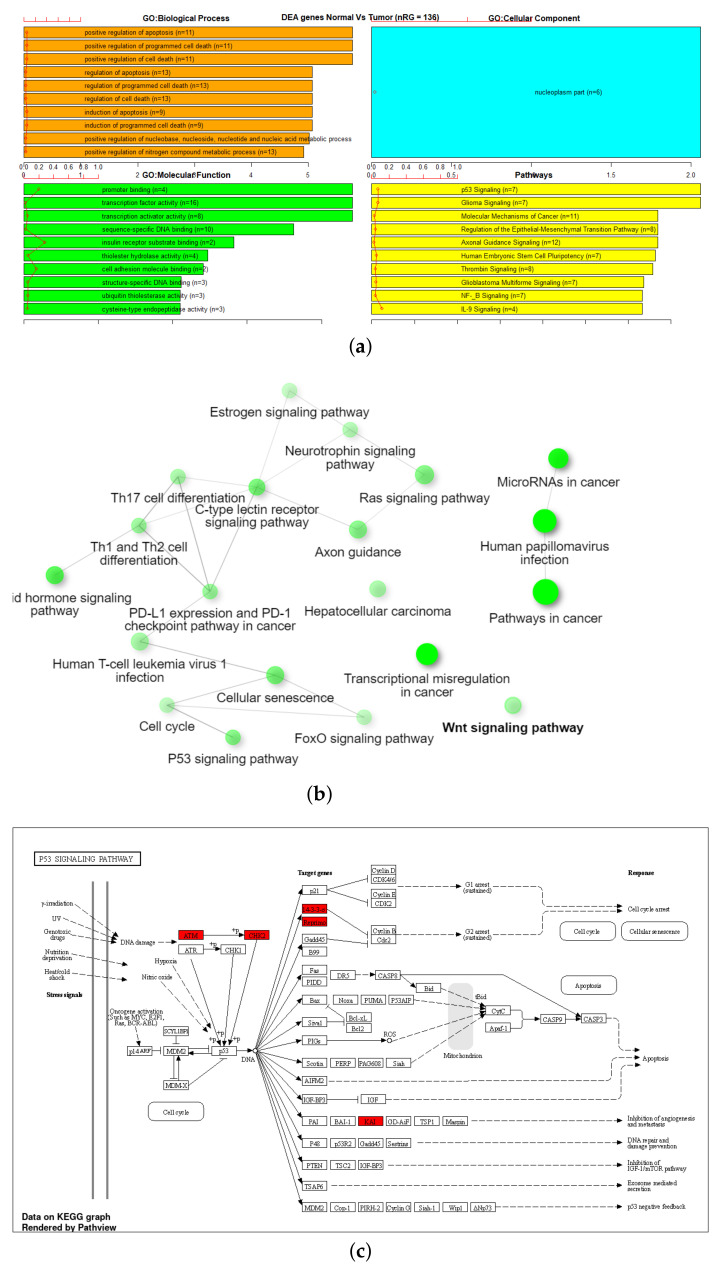
Gene Set Enrichment Analysis on Tumor suppressor/oncogene overlap subset of 450K with SMOTE, (**a**) TCGA EAanalysis where orange represents biological processes, cyan is a cellular component, green is molecular functions, and yellow represents pathways, (**b**) ShinyGO with network visualization of functional associations, and (**c**) ShinyGO enriched pathway visualization where red genes are in the gene set [39,40].

**Figure 10 genes-13-01557-f010:**
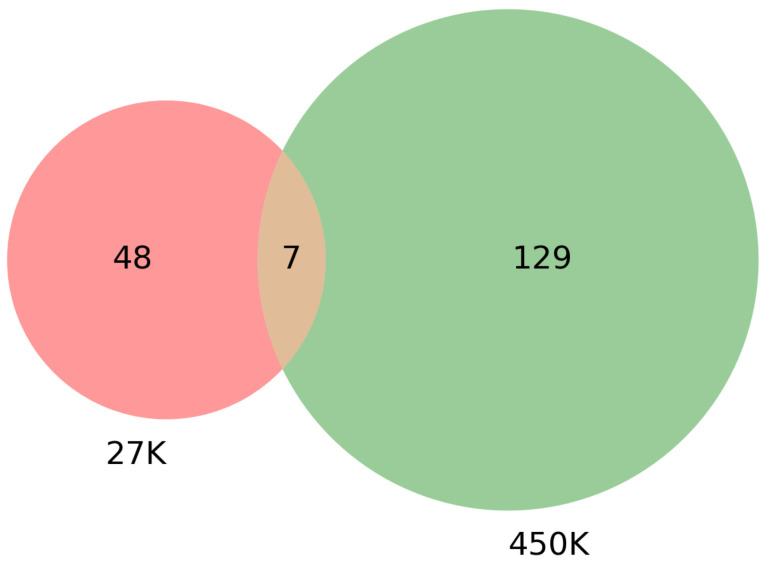
Venn diagram showing the overlap between genes that were found in TSGene + COSMIC [36,37,38] and also found in the 585 out of 685 mapped genes in 27K ANOVA-RF with SMOTE and 1290 out of 1572 mapped genes in 450K ANOVA-RF with SMOTE. A summary of these results can be seen in Table 4. Results indicate that 55 genes were common between TS + COSMIC and 27K while 136 between TS + COSMIC and 450K. There were 7 genes common between all of them.

**Figure 11 genes-13-01557-f011:**
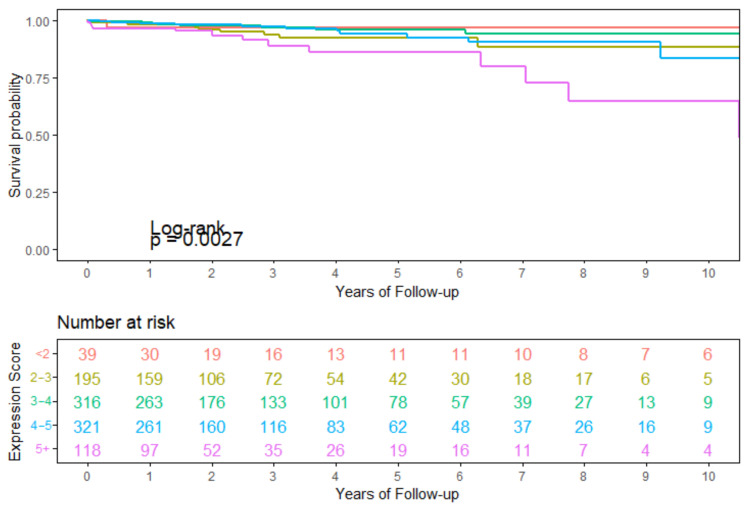
10–year survival using TCGA-BRCA data and an expression score calculated across the seven genes which overlapped between the 27K with SMOTE, 450K with SMOTE, and the tumor suppressor/oncogene lists.

**Table 1 genes-13-01557-t001:** Characterization of CpG markers in the breast cancer datasets.

Dataset	Total Samples	Tumor Samples	Normal Samples	# CpG Markers
27K	337	309	28	27,578
450K	851	750	101	485,577

**Table 2 genes-13-01557-t002:** Mean Squared Error (MSE) and Standard Deviation (STD) for the imputed results using the 27K and 450K datasets. A smaller value of MSE and STD signifies better model imputation.

	Metric	Zero Impute	KNN Impute	Mean Impute	Iterative Impute
**27K**	**MSE**	0.016648	0.016755	0.016749	0.016777
	**STD**	0.007299	0.007253	0.007245	0.007340
**450K**	**MSE**	0.017244	0.017253	0.017251	——–
	**STD**	0.005273	0.005286	0.005307	——–

**Table 3 genes-13-01557-t003:** Model training set-up for 27K and 450K dataset.

	Dataset	# Features	Sample Size	Tumor Samples	Normal Samples	Runtime
**27K**	**All markers**	24,981	337	309	28	21 s
	**Anova_RF**	336	337	309	28	12 s
	**Anova_RF** **(with Smote)**	475	618	309	309	13 s
**450K**	**450K All** **(base + large)**	395,722	851	750	101	1:44:10 s
	**Anova_RF** **(base + large)**	1044	851	750	101	38:41 s
	**Anova_RF** **with SMOTE** **(base + large)**	1445	1500	525	525	13 s

**Table 4 genes-13-01557-t004:** Evaluation of selected oncogenic and tumor suppressor associated gene sets identified to be associated with breast cancer.

Dataset	CpG Markers	Total Genes	COSMIC + TSGene Overlap (3326 Genes)	Sample Genes Overlap (100 Genes)
27K all	24,981	18,166	1214	98
27K ANOVA-RF	336	470	36	2
27K ANOVA-RF SMOTE	475	685	55	6
450K all	395,722	35,555	1455	100
450K ANOVA-RF	1044	1208	88	7
450K ANOVA-RF SMOTE	1445	1572	136	9

## Data Availability

Links to our scripts used for analysis can be found here: https://github.com/rahulgomes19/Deep_Learning_Methylation (accessed on 10 August 2022).

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
