# Peer review of "Application of Feature Selection and Deep Learning for Cancer Prediction Using DNA Methylation Markers"

_genes, 2022, doi:10.3390/genes13091557_

Round 1
Reviewer 1 Report
Title: Application of feature selection and deep learning for cancer prediction using DNA methylation markers
Comments to the Author
The work presented in the manuscript utilized machine learning feature selection methods and deep learning methods, particularly Artificial Neural Network (ANN) methods, to predict the biomarkers for cancer prediction and demonstrated the approach to breast cancer classification. The authors' methods are standard, and the authors show good accuracy. The manuscript is written well. However, the authors need to clarify a few parts, which will also improve the quality of the manuscript.
Major comments
- The abstract is well written but seems very long. Is it possible to reduce? Although I am not sure if the journal has a limitation. Just to be precise, I suggest shortening a bit if possible.
- The introduction section is good. There is a minor concern related to citation on page 2, line 82. The authors can write “In a study by Angermuelle et al.,” instead of “In [22], DNA and CpG modules from Single Cell Bisulfite Sequencing 82 (scBS-seq) data….”
- Regarding the datasets, the authors did not focus on subtypes or grade information; this must be mentioned.
- Right now, the evaluation metrics are provided as bar graphs only. Neither confusion matric nor bar graph is provided. Many important evaluation metrics are missing, and difficult to access the model.
- The authors should provide more details on the figure legend. For example, what does the color means in Figure 6a? Provide network interpretation in figure 6b. The same applies to figure 7.
- The authors mentioned the strengths of the work in the discussion section. However, the authors should provide weaknesses of the study as well.
- Furthermore, the authors could provide the workflow for reproducibility. For example, a GitHub repository. Currently, there is no way to reproduce the work.
- Further, the parameters, package version, etc., must be explained in detail for the same.
- The authors should clarify in a better way the novelty of the work. Also, the authors have made any attempt to check whether these precision markers have any effect on patient survival? Since TCGA got the survival, it can be possible.
- Overall, the authors should provide a vigorous spell check and grammar check, just in case.
Author Response
Thank you for your feedback, the manuscript is much improved.

Reviewer 2 Report
Main comments
1. Abstract should be condensed by about 50%; the first 10 lines can be omitted completely. Please avoid terms that cannot be understood without having read the full paper, such as SMOTE.
2. The introduction is more suited for a review on this topic, please condense and focus on the question that is being addressed in this paper.
3. Line 253-256: “It was observed that the original 450k dataset performed poorly. The reason is probably the excessive number of features that makes it difficult for the simple deep learning architecture to detect meaningful changes. The filtered dataset, despite having a similarly large number of features, performs much more reasonably.” This cannot be gleaned from Table 3. What are “poorly” and “more reasonably” referring to?
4. It was totally unclear to this reviewer how figures 4 and 5 were derived. What is the difference between accuracy, precision, and recall? I suppose it reflects how good the model predicted the sample as normal or as cancer, but it is not explained anywhere in the paper.
5. Par. 3.3 is very hard to follow. There are four sets of CpG markers obtained, but there is no description of these four sets. Then there are six different gene sets, were they derived from the four CpG marker sets? The relevance of the data presentation in figures 6-9 are completely elusive to this reviewer. These kinds of analyses always lead to cancer-relevant gene sets, what’s the novelty? It’s comparable to a weather report in this way.
6. The real critical issue with this paper is the application of SMOTE to pump up the proportion of normal samples in the datasets. The authors show that this leads to higher performance of their models, but how robust are these predictions in new datasets? Without such external replication, the effect of oversampling by SMOTE is hard to gauge and this paper isn’t much more than an advertisement for SMOTE.
Please explain:
1. Line 112: GDC Data Portal (or provide reference)
2. Line 118: these markers were removed or imputed markers before proceeding further. Not clear what exactly was done here, removing or imputing or both?
Typos:
1. Line 65: prostrate
Author Response

(The authors gave the same response as above.)

Round 2
Reviewer 2 Report
Within a couple of days, the authors have addressed all my comments adequately and revised the paper extensively, for which I would like to commend them. However, my main criticism, i.e., lack of external validation of the model with 7 overlapping genes, can obviously not be addressed within a couple of days. I think this significantly reduces the impact of this manuscript.
One could argue that the 27K and 450K datasets are independent, but the Methods section does not provide any details on the cases included besides their numbers. E.g., were the 309 tumor samples in the 27K dataset also analyzed in the 450K dataset?
In addition, because both sets were analyzed in parallel (as far as I could tell), they could serve as each other’s validation, in a way, although the methylation platforms are strongly different. But the analyses aren’t presented in this way and the fact is that the overlap between the two results (figure 10) is almost non-existent. The odds that this is a chance observation is looming large.
Hence the paper should discuss the merits of the datasets in more detail, as well as say something about the robustness of the result. What is the difference between the 27K and 450K in terms of genome representation or informativeness per gene locus? Can this explain the poor overlap? Or is it the sample set itself? The seven genes in the overlap are not major players in breast cancer given what we know today about the molecular genetics of this cancer, and despite the few references listed that claim the opposite: is this result robust?
Finally, the authors have now added a survival analysis, which is commendable. In a way, this adds credibility to the 7-gene set, although I’ve seen similar results in many bioinformatics papers before of which nothing was ever heard of since. In any case, nothing is said about this analysis in the Methods section (how were cases selected for the analysis, what statistic was used, etc)
The log-rank p-value is significant, but as stated by the authors, probably caused mostly by group 5+, which represent only 12% of the patients in the analysis. Since there is hardly a difference between the other groups, these could be lumped, or perhaps there is some other way to dichotomize the data into two groups with distinct methylation features and survival?
Minor details:
Lines 2383-285: “These genes were RTN4IP1, MYO18B, ANP32A, BRF1, SETBP1, NTRK1, IGF2R.” That’s very strange after a sentence mentioning 136 genes. This sentence should be moved to the end of the paragraph at line 308.
Line 345: sentences prematurely broken off.
Author Response
Thank you for the helpful comments!
